# Return-Based Contrastive Representation Learning for Reinforcement Learning

**Guoqing Liu**[1,*]**, Chuheng Zhang**[2,*]**, Li Zhao**[3]**,**
**Tao Qin**[3]**, Jinhua Zhu**[1]**, Jian Li**[2]**, Nenghai Yu**[1]**, Tie-Yan Liu**[3]
[1]University of Science and Technology of China
[2]IIIS, Tsinghua University
[3]Microsoft Research
`lgq1001@mail.ustc.edu.cn, zhangchuheng123@live.com`
`{lizo, taoqin, tyliu}@microsoft.com, teslazhu@mail.ustc.edu.cn`
`lijian83@mail.tsinghua.edu.cn, ynh@ustc.edu.cn`

## Abstract

Recently, various auxiliary tasks have been proposed to accelerate representation learning and improve sample efficiency in deep reinforcement learning (RL). However, existing auxiliary tasks do not take the characteristics of RL problems into consideration and are unsupervised. By leveraging returns, the most important feedback signals in RL, we propose a novel auxiliary task that forces the learnt representations to discriminate state-action pairs with different returns. Our auxiliary loss is theoretically justified to learn representations that capture the structure of a new form of state-action abstraction, under which state-action pairs with similar return distributions are aggregated together. In low data regime, our algorithm outperforms strong baselines on complex tasks in Atari games and DeepMind Control suite, and achieves even better performance when combined with existing auxiliary tasks.

## 1 Introduction

Deep reinforcement learning (RL) algorithms can learn representations from high-dimensional inputs, as well as learn policies based on such representations to maximize long-term returns simultaneously. However, deep RL algorithms typically require large numbers of samples, which can be quite expensive to obtain (Mnih et al., 2015). In contrast, it is usually much more sample efficient to learn policies with learned representations/extracted features (Srinivas et al., 2020). To this end, various auxiliary tasks have been proposed to accelerate representation learning in aid of the main RL task (Suddarth and Kergosien, 1990; Sutton et al., 2011; Gelada et al., 2019; Bellemare et al., 2019; François-Lavet et al., 2019; Shen et al., 2020; Zhang et al., 2020; Dabney et al., 2020; Srinivas et al., 2020). Representative examples of auxiliary tasks include predicting the future in either the pixel space or the latent space with reconstruction-based losses (e.g., Jaderberg et al., 2016; Hafner et al., 2019a;b).

Recently, contrastive learning has been introduced to construct auxiliary tasks and achieves better performance compared to reconstruction based methods in accelerating RL algorithms (Oord et al., 2018; Srinivas et al., 2020). Without the need to reconstruct inputs such as raw pixels, contrastive learning based methods can ignore irrelevant features such as static background in games and learn more compact representations. Oord et al. (2018) propose a contrastive representation learning method based on the temporal structure of state sequence. Srinivas et al. (2020) propose to leverage the prior knowledge from computer vision, learning representations that are invariant to image augmentation. However, existing works mainly construct contrastive auxiliary losses in an unsupervised manner, without considering feedback signals in RL problems as supervision.

In this paper, we take a further step to leverage the return feedback to design a contrastive auxiliary loss to accelerate RL algorithms. Specifically, we propose a novel method, called Return-based

---

*This work is conducted at Microsoft Research Asia. The first two authors contributed equally to this work.

Contrastive representation learning for Reinforcement Learning (RCRL). In our method, given an anchor state-action pair, we choose a state-action pair with the same or similar return as the positive sample, and a state-action pair with different return as the negative sample. Then, we train a discriminator to classify between positive and negative samples given the anchor based on their representations as the auxiliary task. The intuition here is to learn state-action representations that capture return-relevant features while ignoring return-irrelevant features.

From a theoretical perspective, RCRL is supported by a novel state-action abstraction, called $Z^\pi$-irrelevance. $Z^\pi$-irrelevance abstraction aggregates state-action pairs with similar return distributions under certain policy $\pi$. We show that $Z^\pi$-irrelevance abstraction can reduce the size of the state-action space (cf. Appendix A) as well as approximate the Q values arbitrarily accurately (cf. Section 4.1). We further propose a method called Z-learning that can calculate $Z^\pi$-irrelevance abstraction with sampled returns rather than the return distribution, which is hardly available in practice. Z-learning can learn $Z^\pi$-irrelevance abstraction provably efficiently. Our algorithm RCRL can be seen as the empirical version of Z-learning by making a few approximations such as integrating with deep RL algorithms, and collecting positive pairs within a consecutive segment in a trajectory of the anchors.

We conduct experiments on Atari games (Bellemare et al., 2013) and DeepMind Control suite (Tassa et al., 2018) in low data regime. The experiment results show that our auxiliary task combined with Rainbow (Hessel et al., 2017) for discrete control tasks or SAC (Haarnoja et al., 2018) for continuous control tasks achieves superior performance over other state-of-the-art baselines for this regime. Our method can be further combined with existing unsupervised contrastive learning methods to achieve even better performance. We also perform a detailed analysis on how the representation changes during training with/without our auxiliary loss. We find that a good embedding network assigns similar/dissimilar representations to state-action pairs with similar/dissimilar return distributions, and our algorithm can boost such generalization and speed up training.

Our contributions are summarized as follows:

- We introduce a novel contrastive loss based on return, to learn return-relevant representations and speed up deep RL algorithms.
- We theoretically build the connection between the contrastive loss and a new form of state-action abstraction, which can reduce the size of the state-action space as well as approximate the Q values arbitrarily accurately.
- Our algorithm achieves superior performance against strong baselines in Atari games and DeepMind Control suite in low data regime. Besides, the performance can be further enhanced when combined with existing auxiliary tasks.

## 2 RELATED WORK

### 2.1 AUXILIARY TASK

In reinforcement learning, the auxiliary task can be used for both the model-based setting and the model-free setting. In the model-based settings, world models can be used as auxiliary tasks and lead to better performance, such as CRAR (François-Lavet et al., 2019), Dreamer (Hafner et al., 2019a), and PlaNet (Hafner et al., 2019b). Due to the complex components (e.g., the latent transition or reward module) in the world model, such methods are empirically unstable to train and relies on different regularizations to converge. In the model-free settings, many algorithms construct various auxiliary tasks to improve performance, such as predicting the future (Jaderberg et al., 2016; Shelhamer et al., 2016; Guo et al., 2020; Lee et al., 2020; Mazoure et al., 2020), learning value functions with different rewards or under different policies (Veeriah et al., 2019; Schaul et al., 2015; Borsa et al., 2018; Bellemare et al., 2019; Dabney et al., 2020), learning from many-goals (Veeriah et al., 2018), or the combination of different auxiliary objectives (de Bruin et al., 2018). Moreover, auxiliary tasks can be designed based on the prior knowledge about the environment (Mirowski et al., 2016; Shen et al., 2020; van der Pol et al., 2020) or the raw state representation (Srinivas et al., 2020). Hessel et al. (2019) also apply auxiliary task to the multi-task RL setting.

Contrastive learning has seen dramatic progress recently, and been introduced to learn state representation (Oord et al., 2018; Sermanet et al., 2018; Dwibedi et al., 2018; Aytar et al., 2018; Anand

et al., 2019; Srinivas et al., 2020). Temporal structure (Sermanet et al., 2018; Aytar et al., 2018) and local spatial structure (Anand et al., 2019) has been leveraged for state representation learning via contrastive losses. CPC (Oord et al., 2018) and CURL (Srinivas et al., 2020) adopt a contrastive auxiliary tasks to accelerate representation learning and speed up main RL tasks, by leveraging the temporal structure and image augmentation respectively. To the best of our knowledge, we are the first to leverage return to construct a contrastive auxiliary task for speeding up the main RL task.

## 2.2 ABSTRACTION

State abstraction (or state aggregation) aggregates states by ignoring irrelevant state information. By reducing state space, state abstraction can enable efficient policy learning. Different types of abstraction are proposed in literature, ranging from fine-grained to coarse-grained abstraction, each reducing state space to a different extent. Bisimulation or model irrelevance (Dean and Givan, 1997; Givan et al., 2003) define state abstraction under which both transition and reward function are kept invariant. By contrast, other types of state abstraction that are coarser than bisimulation such as $Q^\pi$ irrelevance or $Q^*$ irrelevance (see e.g., Li et al., 2006), which keep the Q function invariant under any policy $\pi$ or the optimal policy respectively. There are also some works on state-action abstractions, e.g., MDP homomorphism (Ravindran, 2003; Ravindran and Barto, 2004a) and approximate MDP homomorphism (Ravindran and Barto, 2004b; Taylor et al., 2009) , which are similar to bisimulation in keeping reward and transition invariant, but extending bisimulation from state abstraction to state-action abstraction.

In this paper, we consider a new form of state-action abstraction $Z^\pi$-irrelevance, which aggregates state-action pairs with the same return distribution and is coarser than bisimulation or homomorphism which are frequently used as auxiliary tasks (e.g., Biza and Platt, 2018; Gelada et al., 2019; Zhang et al., 2020). However, it is worth noting that $Z^\pi$-irrelevance is only used to build the theoretical foundation of our algorithm, and show that our proposed auxiliary task is well-aligned with the main RL task. Representation learning in deep RL is in general very different from aggregating states in tabular case, though the latter may build nice theoretical foundation for the former. Here we focus on how to design auxiliary tasks to accelerate representation learning using contrastive learning techniques, and we propose a novel return-based contrastive method based on our proposed $Z^\pi$-irrelevance abstraction.

## 3 PRELIMINARY

We consider a Markov Decision Process (MDP) which is a tuple $(\mathcal{S}, \mathcal{A}, P, R, \mu, \gamma)$ specifying the state space $\mathcal{S}$, the action space $\mathcal{A}$, the state transition probability $P(s_{t+1}|s_t, a_t)$, the reward $R(r_t|s_t, a_t)$, the initial state distribution $\mu \in \Delta^{\mathcal{S}}$ and the discount factor $\gamma$. Also, we denote $x := (s, a) \in \mathcal{X} := \mathcal{S} \times \mathcal{A}$ to be the state-action pair. A (stationary) policy $\pi : \mathcal{S} \to \Delta^{\mathcal{A}}$ specifies the action selection probability on each state. Following the policy $\pi$, the discounted sum of future rewards (or return) is denoted by the random variable $Z^\pi(s, a) = \sum_{t=0}^{\infty} \gamma^t R(s_t, a_t)$, where $s_0 = s, a_0 = a, s_t \sim P(\cdot|s_{t-1}, a_{t-1})$, and $a_t \sim \pi(\cdot|s_t)$. We divide the range of return into $K$ equal bins $\{R_0 = R_{\min}, R_1, \cdots, R_K = R_{\max}\}$ such that $R_k - R_{k-1} = (R_{\max} - R_{\min})/K, \forall k \in [K]$, where $R_{\min}$ (resp. $R_{\max}$) is the minimum (reps. maximum) possible return, and $[K] := \{1, 2, \cdots, K\}$. We use $b(R) = k \in [K]$ to denote the event that $R$ falls into the $k$th bin, i.e., $R_{k-1} < R \le R_k$. Hence, $b(R)$ can be viewed as the discretized version of the return, and the distribution of discretized return can be represented by a $K$-dimensional vector $\mathbb{Z}^\pi(x) \in \Delta^K$, where the $k$-th element equals to $\Pr[R_{k-1} < Z^\pi(x) \le R_k]$. The Q function is defined as $Q^\pi(x) = \mathbb{E}[Z^\pi(x)]$, and the state value function is defined as $V^\pi(s) = \mathbb{E}_{a \sim \pi(\cdot|s)}[Z^\pi(s, a)]$. The objective for RL is to find a policy $\pi$ that maximizes the expected cumulative reward $J(\pi) = \mathbb{E}_{s \sim \mu}[V^\pi(s)]$. We denote the optimal policy as $\pi^*$ and the corresponding optimal Q function as $Q^* := Q^{\pi^*}$.

## 4 METHODOLOGY

In this section, we present our method, from both theoretical and empirical perspectives. First, we propose $Z^\pi$-irrelevance, a new form of state-action abstraction based on return distribution. We show that the Q functions for any policy (and therefore the optimal policy) can be represented under

---

**Algorithm 1:** Z-learning

---

1: Given the policy $\pi$, the number of bins for the return $K$, a constant $N \geq N_{\pi,K}$, the encoder class $\Phi_N$, the regressor class $\mathcal{W}_N$, and a distribution $d \in \Delta^{\mathcal{X}}$ with $\text{supp}(d) = \mathcal{X}$
2: $\mathcal{D} = \emptyset$
3: **for** $i = 1, \cdots, n$ **do**
4:     $x_1, x_2 \sim d$
5:     $R_1 \sim Z^{\pi}(x_1), R_2 \sim Z^{\pi}(x_2)$
6:     $\mathcal{D} = \mathcal{D} \cup \{(x_1, x_2, y = \mathbb{I}[b(R_1) \neq b(R_2)])\}$
7: **end for**
8: $(\hat{\phi}, \hat{w}) = \arg\min_{\phi \in \Phi_N, w \in \mathcal{W}_N} \mathcal{L}(\phi, w; \mathcal{D})$, where $\mathcal{L}(\phi, w; \mathcal{D})$ is defined in (1)
9: **return** the encoder $\hat{\phi}$

---

$Z^{\pi}$-irrelevance abstraction. Then we consider an algorithm, Z-learning, that enables us to learn $Z^{\pi}$-irrelevance abstraction from the samples collected using $\pi$. Z-learning is simple and learns the abstraction by only minimizing a contrastive loss. We show that Z-learning can learn $Z^{\pi}$-irrelevance abstraction provably efficiently. After that, we introduce return-based contrastive representation learning for RL (RCRL) that incorporates standard RL algorithms with an auxiliary task adapted from Z-learning. At last, we present our network structure for learning state-action embedding, upon which RCRL is built.

## 4.1 $Z^{\pi}$-IRRELEVANCE ABSTRACTION

A state-action abstraction aggregates the state-action pairs with similar properties, resulting in an abstract state-action space denoted as $[N]$, where $N$ is the size of abstract state-action space. In this paper, we consider a new form of abstraction, $Z^{\pi}$-irrelevance, defined as follows: Given a policy $\pi$, $Z^{\pi}$-irrelevance abstraction is denoted as $\phi : \mathcal{X} \to [N]$ such that, for any $x_1, x_2 \in \mathcal{X}$ with $\phi(x_1) = \phi(x_2)$, we have $\mathbb{Z}^{\pi}(x_1) = \mathbb{Z}^{\pi}(x_2)$. Given a policy $\pi$ and the parameter for return discretization $K$, we use $N_{\pi,K}$ to denote the minimum $N$ such that a $Z^{\pi}$-irrelevance exists. It is true that $N_{\pi,K} \leq N_{\pi,\infty} \leq |\phi_B(\mathcal{S})| |\mathcal{A}|$ for any $\pi$ and $K$, where $|\phi_B(\mathcal{S})|$ is the number of abstract states for the coarsest bisimulation (cf. Appendix A).

**Proposition 4.1.** *Given a policy $\pi$ and any $Z^{\pi}$-irrelevance $\phi : \mathcal{X} \to [N]$, there exists a function $Q : [N] \to \mathbb{R}$ such that $|Q(\phi(x)) - Q^{\pi}(x)| \leq \frac{R_{\max} - R_{\min}}{K}, \forall x \in \mathcal{X}$.*

We provide a proof in Appendix A. Note that $K$ controls the coarseness of the abstraction. When $K \to \infty$, $Z^{\pi}$-irrelevance can accurately represent the value function and therefore the optimal policy when $\pi \to \pi^*$. When using an auxiliary task to learn such abstraction, this proposition indicates that the auxiliary task (to learn a $Z^{\pi}$-irrelevance) is well-aligned with the main RL task (to approximate $Q^*$). However, large $K$ results in a fine-grained abstraction which requires us to use a large $N$ and more samples to learn the abstraction (cf. Theorem 4.1). In practice, this may not be a problem since we learn a state-action representation in a low-dimensional space $\mathbb{R}^d$ instead of $[N]$ and reuse the samples collected by the base RL algorithm. Also, we do not need to choose a $K$ explicitly in the practical algorithm (cf. Section 4.3).

## 4.2 Z-LEARNING

We propose Z-learning to learn $Z^{\pi}$-irrelevance based on a dataset $\mathcal{D}$ with a contrastive loss (see Algorithm 1). Each tuple in the dataset is collected as follows: First, two state-action pairs are drawn i.i.d. from a distribution $d \in \Delta^{\mathcal{X}}$ with $\text{supp}(d) = \mathcal{X}$ (cf. Line 4 in Algorithm 1). In practice, we can sample state-action pairs from the rollouts generated by the policy $\pi$. In this case, a stochastic policy (e.g., using $\epsilon$-greedy) with a standard ergodic assumption on MDP ensures $\text{supp}(d) = \mathcal{X}$. Then, we obtain returns for the two state-action pairs (i.e., the discounted sum of the rewards after $x_1$ and $x_2$) which can be obtained by rolling out using the policy $\pi$ (cf. Line 5 in Algorithm 1). The binary label $y$ for this state-action pair indicates whether the two returns belong to the same bin (cf. Line 6 in Algorithm 1). The contrastive loss is defined as follows:

$$\min_{\phi \in \Phi_N, w \in \mathcal{W}_N} \mathcal{L}(\phi, w; \mathcal{D}) := \mathbb{E}_{(x_1, x_2, y) \sim \mathcal{D}} \left[ (w(\phi(x_1), \phi(x_2)) - y)^2 \right], \tag{1}$$

where the class of encoders that map the state-action pairs to $N$ discrete abstractions is defined as $\Phi_N := \{\mathcal{X} \rightarrow [N]\}$, and the class of tabular regressors is defined as $\mathcal{W}_N := \{[N] \times [N] \rightarrow [0,1]\}$. Notice that we choose $N \geq N_{\pi,K}$ to ensure that a $Z^\pi$-irrelevance $\phi : \mathcal{X} \rightarrow [N]$ exists. Also, to aggregate the state-action pairs, $N$ should be smaller than $|\mathcal{X}|$ (otherwise we will obtain an identity mapping). In this case, mapping two state-action pairs with different return distributions to the same abstraction will increase the loss and therefore is avoided. The following theorem shows that Z-learning can learn $Z^\pi$-irrelevance provably efficiently.

**Theorem 4.1.** *Given the encoder $\hat{\phi}$ returned by Algorithm 1, the following inequality holds with probability $1 - \delta$ and for any $x' \in \mathcal{X}$:*

$$
\mathbb{E}_{x_1 \sim d, x_2 \sim d}\Big[\mathbb{I}[\hat{\phi}(x_1) = \hat{\phi}(x_2)] \Big| \mathbb{Z}^\pi(x')^T \left(\mathbb{Z}^\pi(x_1) - \mathbb{Z}^\pi(x_2)\right) \Big|\Big]
$$
$$
\leq \sqrt{\frac{8N}{n}\Big(3 + 4N^2 \ln n + 4\ln|\Phi_N| + 4\ln(\frac{2}{\delta})\Big)},
$$
(2)

*where $|\Phi_N|$ is the cardinality of encoder function class and $n$ is the size of the dataset.*

We provide the proof in Appendix B. Although $|\Phi_N|$ is upper bounded by $N^{|\mathcal{X}|}$, it is generally much smaller for deep encoders that generalize over the state-action space. The theorem shows that whenever $\hat{\phi}$ maps two state-actions $x_1, x_2$ to the same abstraction, $\mathbb{Z}^\pi(x_1) \approx \mathbb{Z}^\pi(x_2)$ up to an error proportional to $1/\sqrt{n}$ (ignoring the logarithm factor). The following corollary shows that $\hat{\phi}$ becomes a $Z^\pi$-irrelevance when $n \rightarrow \infty$.

**Corollary 4.1.1.** *The encoder $\hat{\phi}$ returned by Algorithm 1 with $n \rightarrow \infty$ is a $Z^\pi$-irrelevance, i.e., for any $x_1, x_2 \in \mathcal{X}$, $\mathbb{Z}^\pi(x_1) = \mathbb{Z}^\pi(x_2)$ if $\hat{\phi}(x_1) = \hat{\phi}(x_2)$.*

### 4.3 RETURN-BASED CONTRASTIVE LEARNING FOR RL (RCRL)

We adapt Z-learning as an auxiliary task that helps the agent to learn a representation with meaningful semantics. The auxiliary task based RL algorithm is called RCRL and shown in Algorithm 2. Here, we use Rainbow (for discrete control) and SAC (for continuous control) as the base RL algorithm for RCRL. However, RCRL can also be easily incorporated with other model-free RL algorithms. While Z-learning relies on a dataset sampled by rolling out the current policy, RCRL constructs such a dataset using the samples collected by the base RL algorithm and therefore does not require additional samples, e.g., directly using the replay buffer in Rainbow or SAC (see Line 7 and 8 in Algorithm 2). Compared with Z-learning, we use the state-action embedding network that is shared with the base RL algorithm $\phi : \mathcal{X} \rightarrow \mathbb{R}^d$ as the encoder, and use an additional discriminator trained by the auxiliary task $w : \mathbb{R}^d \times \mathbb{R}^d \rightarrow [0,1]$ as the regressor.

However, when implementing Z-learning as the auxiliary task, the labels in the dataset may be unbalanced. Although this does not cause problems in the theoretical analysis since we assume the Bayes optimizer can be obtained for the contrastive loss, it may prevent the discriminator from learning properly in practice (cf. Line 8 in Algorithm 1). To solve this problem, instead of drawing samples independently from the replay buffer $\mathcal{B}$ (analogous to sampling from the distribution $d$ in Z-learning), we sample the pairs for $\mathcal{D}$ as follows: As a preparation, we cut the trajectories in $\mathcal{B}$ into segments, where each segment contains state-action pairs with the same or similar returns. Specifically, in Atari games, we create a new segment once the agent receives a non-zero reward. In DMControl tasks, we first prescribe a threshold and then create a new segment once the cumulative reward within the current segment exceeds this threshold. For each sample in $\mathcal{D}$, we first draw an anchor state-action pair from $\mathcal{B}$ randomly. Afterwards, we generate a positive sample by drawing a state-action pair from the same segment of the anchor state-action pair. Then, we draw another state-action pair randomly from $\mathcal{B}$ and use it as the negative sample.

We believe our auxiliary task may boost the learning, due to better return-induced representations that facilitate generalization across different state-action pairs. Learning on one state-action pair will affect the value of all state-action pairs that share similar representations with this pair. When the embedding network assigns similar representations to similar state-action pairs (e.g., sharing similar distribution over returns), the update for one state-action pair is representative for the updates for other similar state-action pairs, which improves sample efficiency. However, such generalization may not be achieved by the base RL algorithm since, when trained by the algorithm with only a

---

**Algorithm 2:** Return based Contrastive learning for RL (RCRL)

1 Initialize the embedding $\phi_\theta : \mathcal{X} \rightarrow \mathbb{R}^d$ and a discriminator $w_\vartheta : \mathbb{R}^d \times \mathbb{R}^d \rightarrow [0, 1]$
2 Initialize the parameters $\varphi$ for the base RL algorithm that uses the learned embedding $\phi_\theta$
3 Given a batch of samples $\mathcal{D}$, the loss function for the base RL algorithm is $\mathcal{L}_{\mathrm{RL}}(\phi_\theta, \varphi; \mathcal{D})$
4 A replay buffer $\mathcal{B} = \emptyset$
5 **foreach** *iteration* **do**
6     Rollout the current policy and store the samples to the replay buffer $\mathcal{B}$
7     Draw a batch of samples $\mathcal{D}$ from the replay buffer $\mathcal{B}$
8     Update the parameters with the loss function $\mathcal{L}(\phi_\theta, w_\vartheta; \mathcal{D}) + \mathcal{L}_{\mathrm{RL}}(\phi_\theta, \varphi; \mathcal{D})$
9 **end**
10 **return** *The learned policy*

---

return-based loss, similar state-action pairs may have similar Q values but very different representations. One may argue that, we can adopt several temporal difference updates to propagate the values for state-action pairs with same sampled return, and finally all such pairs are assigned with similar Q values. However, since we adopt a learning algorithm with bootstrapping/temporal difference learning and frozen target network in deep RL, it could take longer time to propagate the value across different state-action pairs, compared with direct generalization over state-action pairs with contrastive learning.

Meanwhile, since we construct auxiliary tasks based on return, which is a very different structure from image augmentation or temporal structure, our method could be combined with existing methods to achieve further improvement.

### 4.4 NETWORK STRUCTURE FOR STATE-ACTION EMBEDDING

In our algorithm, the auxiliary task is based on the state-action embedding, instead of the state embedding that is frequently used in the previous work (e.g., Srinivas et al., 2020). To facilitate our algorithm, we design two new structures for Atari 2600 Games (discrete action) and DMControl Suite (continuous action) respectively. We show the structure in Figure 1. For Atari, we learn an action embedding for each action and use the element-wise product of the state embedding and action embedding as the state-action embedding. For DMControl, the action embedding is a real-valued vector and we use the concatenation of the action embedding and the state embedding.

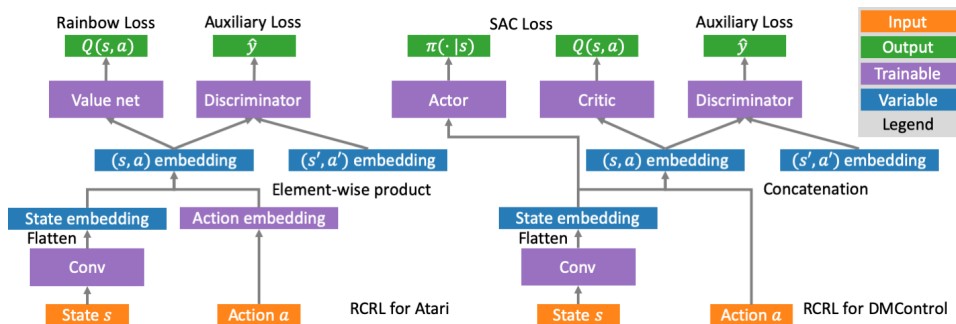

Figure 1: The network structure of RCRL for Atari (left) and DMControl Suite (right).

## 5 EXPERIMENT

In this section, we conduct the following experiments: 1) We implement RCRL on Atari 2600 Games (Bellemare et al., 2013) and DMControl Suite (Tassa et al., 2020), and compare with state-of-the-art model-free methods and strong model-based methods. In particular, we compare with CURL (Srinivas et al., 2020), a top performing auxiliary task based RL algorithm for pixel-based control tasks that also uses a contrastive loss. In addition, we also combine RCRL with CURL to study whether our auxiliary task further boosts the learning when combined with other auxiliary

tasks. 2) To further study the reason why our algorithm works, we analyze the generalization of the learned representation. Specifically, we compare the cosine similarity between the representations of different state-action pairs. We provide the implementation details in Appendix C.

## 5.1 EVALUATION ON ATARI AND DMCONTROL

**Experiments on Atari.** Our experiments on Atari are conducted in low data regime of 100k interactions between the agent and the environment, which corresponds to two hours of human play. We show the performance of different algorithms/baselines, including the scores for average human (Human), SimPLe (Kaiser et al., 2019) which is a strong model-based baseline for this regime, original Rainbow (Hessel et al., 2017), Data-Efficient Rainbow (ERainbow, van Hasselt et al., 2019), ERainbow with state-action embeddings (ERainbow-sa, cf. Figure 1 Left), CURL (Srinivas et al., 2020) that is based on ERainbow, RCRL which is based on ERainbow-sa, and the algorithm that combines the auxiliary loss for CURL and RCRL to ERainbow-sa (RCRL+CURL). We show the evaluation results of our algorithm and the baselines on Atari games in Table 1. First, we observe that using state-action embedding instead of state embedding in ERainbow does not lead to significant performance change by comparing ERainbow with ERainbow-sa. Second, built upon ERainbow-sa, the auxiliary task in RCRL leads to better performance compared with not only the base RL algorithm but also SimPLe and CURL in terms of the median human normalized score (HNS). Third, we can see that RCRL further boosts the learning when combined with CURL and achieves the best performance for 7 out of 26 games, which shows that our auxiliary task can be successfully combined with other auxiliary tasks that embed different sources of information to learn the representation.

| | Human | SimPLe | Rainbow | ERainbow | ERainbow-sa | CURL | RCRL | RCRL+CURL |
|---|---|---|---|---|---|---|---|---|
| ALIEN | 7127.7 | 616.9 | 318.7 | 739.9 | 813.8 | 558.2 | 854.2 | **912.2** |
| AMIDAR | 1719.5 | 88.0 | 32.5 | **188.6** | 154.2 | 142.1 | 157.7 | 125.1 |
| ASSAULT | 742.0 | 527.2 | 231.0 | 431.2 | 576.2 | **600.6** | 569.6 | 588.4 |
| ASTERIX | 8503.3 | **1128.3** | 243.6 | 470.8 | 697.0 | 734.5 | 799.0 | 683.0 |
| BANK HEIST | 753.1 | 34.2 | 15.6 | 51.0 | 96.0 | **131.6** | 107.2 | 99.0 |
| BATTLE ZONE | 37187.5 | 5184.4 | 2360.0 | 10124.6 | 13920.0 | 14870.0 | 14280.0 | **17380.0** |
| BOXING | 12.1 | **9.1** | -24.8 | 0.2 | 2.2 | 1.2 | 2.7 | 6.7 |
| BREAKOUT | 30.5 | **16.4** | 1.2 | 1.9 | 3.4 | 4.9 | 4.3 | 4.0 |
| CHOPPER COMMAND | 7387.8 | 1246.9 | 120.0 | 861.8 | 1064.0 | 1058.5 | **1262.0** | 1008.0 |
| CRAZY CLIMBER | 35829.4 | **62583.6** | 2254.5 | 16185.3 | 21840.0 | 12146.5 | 15120.0 | 15032.0 |
| DEMON ATTACK | 1971.0 | 208.1 | 163.6 | 508.0 | 768.0 | **817.6** | 790.4 | 618.3 |
| FREEWAY | 29.6 | 20.3 | 0.0 | **27.9** | 26.5 | 26.7 | 26.6 | 25.4 |
| FROSTBITE | 4334.7 | 254.7 | 60.2 | 866.8 | 1472.0 | 1181.3 | 1337.6 | **1516.6** |
| GOPHER | 2412.5 | **771.0** | 431.2 | 349.5 | 384.8 | 669.3 | 429.6 | 458.8 |
| HERO | 30826.4 | 2656.6 | 487.0 | 6857.0 | 4787.9 | 6279.3 | 6454.1 | **7647.4** |
| JAMESBOND | 302.8 | 125.3 | 47.4 | 301.6 | 308.0 | 471.0 | 314.0 | **503.0** |
| KANGAROO | 3035.0 | 323.1 | 0.0 | 779.3 | 732.0 | 872.5 | 842.0 | **932.0** |
| KRULL | 2665.5 | **4539.9** | 1468.0 | 2851.5 | 2740.0 | 4229.6 | 2997.5 | 3905.8 |
| KUNG FU MASTER | 22736.3 | **17257.2** | 0.0 | 14346.1 | 11914.0 | 14307.8 | 9762.0 | 11856.0 |
| MS PACMAN | 6951.6 | 1480.0 | 67.0 | 1204.1 | 1384.5 | 1465.5 | **1555.2** | 1336.8 |
| PONG | 14.6 | **12.8** | -20.6 | -19.3 | -18.3 | -16.5 | -16.9 | -18.72 |
| PRIVATE EYE | 69571.3 | 58.3 | 0.0 | 97.8 | 80.0 | 218.4 | 102.6 | **282.3** |
| QBERT | 13455.0 | **1288.8** | 123.5 | 1152.9 | 893.5 | 1042.4 | 1121.0 | 942.0 |
| ROAD RUNNER | 7845.0 | 5640.6 | 1588.5 | **9600.0** | 5392.0 | 5661.0 | 6138.0 | 5392.0 |
| SEAQUEST | 42054.7 | **683.3** | 131.7 | 354.1 | 402.0 | 384.5 | 375.6 | 489.6 |
| UP N DOWN | 11693.2 | 3350.3 | 504.6 | 2877.4 | 3235.2 | 2955.2 | **4210.2** | 3127.8 |
| Median HNS | 100.0% | 14.4% | 0.0% | 16.1% | 16.7% | 17.5% | 18.5% | **19.6%** |

Table 1: Scores of different algorithms/baselines on 26 games for Atari-100k benchmark. We show the mean score averaged over five random seeds.

**Experiments on DMControl.** For DMControl, we compare our algorithm with the following baselines: Pixel SAC (Haarnoja et al., 2018) which is the base RL algorithm that receives images as the input; SLAC (Lee et al., 2019) that learns a latent variable model and then updates the actor and critic based on it; SAC+AE (Yarats et al., 2019) that uses a regularized autoencoder for reconstruction in the auxiliary task; PlaNet (Hafner et al., 2019b) and Dreamer (Hafner et al., 2019a) that learn a latent space world model and explicitly plan with the learned model. We also compare with a skyline, State SAC, the receives the low-dimensional state representation instead of the image as the input. Different from Atari games, tasks in DMControl yield dense reward. Consequently, we split the trajectories into segments using a threshold such that the difference of returns within each segment does not exceed this threshold. Similarly, we test the algorithms in low data regime of 500k interactions. We show the evaluation results in Figure 2. We can see that our auxiliary task not only brings performance improvement over the base RL algorithm but also outperforms CURL and other state-of-the-art baseline algorithms in different tasks. Moreover, we observe that our algorithm is

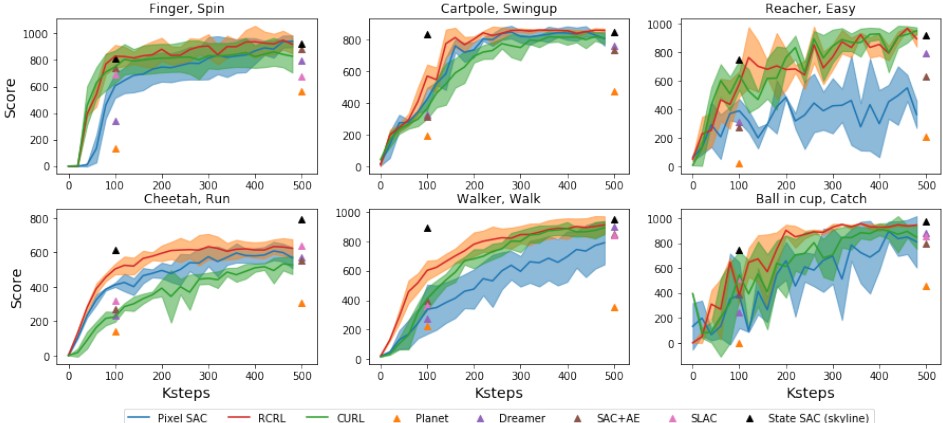

Figure 2: Scores achieved by RCRL and other baseline algorithms during the training for different tasks in DMControl suite. The line and the shaded area indicate the average and the standard deviation over 5 random seeds respectively.

more robust across runs with different seeds compared with Pixel SAC and CURL (e.g., for the task *Ball in cup, Catch*).

## 5.2 ANALYSIS ON THE LEARNED REPRESENTATION

We analyze on the learned representation of our model to demonstrate that our auxiliary task attains a representation with better generalization, which may explain why our algorithm succeeds. We use cosine similarity to measure the generalization from one state-action pair to another in the deep learning model. Given two state-action pairs $x_1, x_2 \in \mathcal{X}$, cosine similarity is defined as $\frac{\phi_\theta(x_1)^T \phi_\theta(x_2)}{||\phi_\theta(x_1)|| \, ||\phi_\theta(x_2)||}$, where $\phi_\theta(\cdot)$ is the learnt embedding network.

We show the cosine similarity of the representations between positive pairs (that are sampled within the same segment and therefore likely to share similar return distributions) and negative pairs (i.e., randomly sampled state-action pairs) during the training on the game *Alien* in Figure 4. First, we observe that when a good policy is learned, the representations of positive pairs are similar while those of negative pairs are dissimilar. This indicates that a good representation (or the representation that supports a good policy) aggregates the state-action pairs with similar return distributions. Then, we find that our auxiliary loss can accelerate such generalization for the representation, which makes RCRL learn faster.

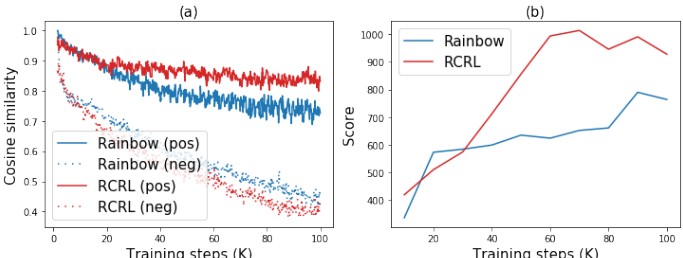

Figure 3: Analysis of the learned representation on *Alien*. (a) The cosine similarity between the representations of the positive/negative state-action pair and the anchor during the training of Rainbow and RCRL. (b) The scores of the two algorithms during the training.

## 6 CONCLUSION

In this paper, we propose return-based contrastive representation learning for RL (RCRL), which introduces a return-based auxiliary task to facilitate policy training with standard RL algorithms. Our auxiliary task is theoretically justified to learn representations that capture the structure of $Z^\pi$-irrelevance, which can reduce the size of the state-action space as well as approximate the Q values arbitrarily accurately. Experiments on Atari games and the DMControl suite in low data regime demonstrate that our algorithm achieves superior performance not only when using our auxiliary task alone but also when combined with other auxiliary tasks, .

As for future work, we are interested in how to combine different auxiliary tasks in a more sophisticated way, perhaps with a meta-controller. Another potential direction would be providing a theoretical analysis for auxiliary tasks and justifying why existing auxiliary tasks can speed up deep RL algorithms.

## 7 ACKNOWLEDGMENT

Guoqing Liu and Nenghai Yu are supported in part by the Natural Science Foundation of China under Grant U20B2047, Exploration Fund Project of University of Science and Technology of China under Grant YD3480002001. Chuheng Zhang and Jian Li are supported in part by the National Natural Science Foundation of China Grant 61822203, 61772297, 61632016 and the Zhongguancun Haihua Institute for Frontier Information Technology, Turing AI Institute of Nanjing and Xi'an Institute for Interdisciplinary Information Core Technology.

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

## A  $Z^\pi$-IRRELEVANCE

### A.1  COMPARISON WITH BISIMULATION.

We consider a bisimulation $\phi$ and denote the set of abstract states as $\mathcal{Z} := \phi(\mathcal{S})$. The bisimulation $\phi_B$ is defined as follows:

**Definition A.1** (Bisimulation (Givan et al., 2003)). *$\phi_B$ is bisimulation if $\forall s_1, s_2 \in \mathcal{S}$ where $\phi_B(s_1) = \phi_B(s_2)$, $\forall a \in \mathcal{A}, z' \in \mathcal{Z}$,*

$$R(s_1, a) = R(s_2, a), \qquad \sum_{s' \in \phi_B^{-1}(z')} P(s'|s_1, a) = \sum_{s' \in \phi_B^{-1}(z')} P(s'|s_2, a).$$

Notice that the coarseness of $Z^\pi$-irrelevance is dependent on $K$, the number of bins for the return. When $K \to \infty$, two state-action pairs $x$ and $x'$ are aggregated only when $Z^\pi(x) \overset{D}{=} Z^\pi(x')$, which is a strict condition resulting in a fine-grained abstraction. Here, we provide the following proposition to illustrate that $Z^\pi$-irrelevance abstraction is coarser than bisimulation even when $K \to \infty$.

**Proposition A.1.** *Given $\phi_B$ to be the coarsest bisimulation, $\phi_B$ induces a $Z^\pi$-irrelevance abstraction for any policy $\pi$ defined over $\mathcal{Z}$. Specifically, if $\forall s_1, s_2 \in \mathcal{S}$ where $\phi_B(s_1) = \phi_B(s_2)$, then $Z^\pi(s_1, a) \overset{D}{=} Z^\pi(s_2, a), \forall a \in \mathcal{A}$ for any policy $\pi$ defined over $\mathcal{Z}$.*

Consider a state-action abstraction $\tilde{\phi}_B$ that is augmented from the coarsest bisimulation $\phi_B$: $\tilde{\phi}_B(s_1, a_1) = \tilde{\phi}_B(s_2, a_2)$ if and only if $\phi_B(s_1) = \phi(s_2)$ and $a_1 = a_2$. The proposition indicates that $|\phi_B(\mathcal{S})||\mathcal{A}| = |\tilde{\phi}_B(\mathcal{X})| \geq N_{\pi,\infty} \geq N_{\pi,K}$ for any $K$ and for any $\pi$ defined over $\mathcal{Z}$, i.e., bisimulation is no coarser than $Z^\pi$-irrelevance. Note that there exists an optimal policy that is defined over $\mathcal{Z}$ (Li et al., 2006). In practice, $Z^\pi$-irrelevance should be far more coarser than bisimulation when we only consider one specific policy $\pi$. Therefore learning a $Z^\pi$-irrelevance should be easier than learning a bisimulation.

*Proof.* First, for a fixed policy $\pi : \mathcal{Z} \to \Delta^{\mathcal{A}}$, we prove that if two state distributions projected to $\mathcal{Z}$ are identical, the corresponding reward distributions or the next state distributions projected to $\mathcal{Z}$ will be identical. Then, we use such invariance property to prove the proposition.

The proof for the invariance property goes as follows: Consider two identical state distributions over $\mathcal{Z}$ on the $t$-th step, $P_1, P_2 \in \Delta^{\mathcal{Z}}$ such that $P_1 = P_2$. Notice that we only require that the two state distributions are identical when projected to $\mathcal{Z}$, so therefore they may be different on $\mathcal{S}$. Specifically, if we denote the state distribution over $\mathcal{S}$ as $P(s) = P(z)q(s|z)$ where $\phi_B(s) = z$, the distribution $q$ for the two state distributions may be different. However, we will show that this is sufficient to ensure that the corresponding state distributions on $\mathcal{Z}$ are identical on the next step (and therefore the subsequent steps).

We denote $R_1$ and $R_2$ as the reward on the $t$-th step, which are random variables. The reward distribution on the $t$-th step is specified as follows:

$$\begin{aligned}
P(R_1 = r|P_1) &= \sum_{z \in \mathcal{Z}, a \in \mathcal{A}, s \in \mathcal{S}} P(r|s, a)q_1(s|z)\pi(a|z)P_1(z) \\
&= \sum_{z \in \mathcal{Z}, a \in \mathcal{A}} P(r|z, a)\pi(a|z)P_1(z) \\
&= \sum_{z \in \mathcal{Z}, a \in \mathcal{A}} P(r|z, a)\pi(a|z)P_2(z) \\
&= P(R_2 = r|P_2),
\end{aligned} \tag{3}$$

for any $r \in \mathbb{R}$, where we use the property of bisimulation, $R|s, a \overset{D}{=} R|s', a, \forall \phi_B(s) = \phi_B(s')$, in the second equation. This indicates that $R_1|P_1 \overset{D}{=} R_2|P_2$.

Similarly, we denote $P_1'$ and $P_2'$ as the state distribution over $\mathcal{Z}$ on the $(t+1)$-th step. We have

$$
\begin{aligned}
P_1'(z')|P_1 &= \sum_{z \in \mathcal{Z}, a \in \mathcal{A}} \sum_{s \in \mathcal{S}} \left( \sum_{s' \in \phi_B^{-1}(z')} P(s'|s, a) \right) q_1(s|z)\pi(a|z)P_1(z) \\
&= \sum_{z \in \mathcal{Z}, a \in \mathcal{A}} P(z'|z, a)\pi(a|z)P_1(z) \\
&= \sum_{z \in \mathcal{Z}, a \in \mathcal{A}} P(z'|z, a)\pi(a|z)P_2(z) \\
&= P_2'(z')|P_1,
\end{aligned}
\tag{4}
$$

for any $z' \in \mathcal{Z}$, where we also use the property of bisimulation in the second equation, i.e., the quantity in the bracket equals for different $s$ such that $q(s|z) > 0$ with a fixed $z$. This indicates that $P_1'|P_1 = P_2'|P_2$.

At last, consider two state $s_1, s_2 \in \mathcal{S}$ with $\phi_B(s_1) = \phi_B(s_2) = z$ on the first step, and we take $a \in \mathcal{A}$ for both states on this step and later take the actions following $\pi$. We denote the subsequent reward and state distribution over $\mathcal{Z}$ as $R_1^{(1)}, P_1^{(2)}, R_1^{(2)}, P_1^{(3)}, \cdots$ and $R_2^{(1)}, P_2^{(2)}, R_2^{(2)}, P_2^{(3)}, \cdots$ respectively, where the superscripts indicate time steps. Therefore, we can deduce that $P_1^{(1)} = P_2^{(1)}, R_1^{(1)} \overset{D}{=} R_2^{(1)}, P_1^{(2)} = P_2^{(2)}, R_1^{(2)} \overset{D}{=} R_2^{(2)}, \cdots$ and therefore $Z^\pi(s_1, a) \overset{D}{=} Z^\pi(s_2, a), \forall a \in \mathcal{A}$.

$\square$

## A.2 Comparison with $\pi$-bisimulation

Similar to $Z^\pi$-irrelevance, a recently proposed state abstraction $\pi$-bisimulation (Castro, 2020) is also tied to a behavioral policy $\pi$. It is beneficial to compare the coarseness of $Z^\pi$-irrelevance and $\pi$-bisimulation. For completeness, we restate the definition of $\pi$-bisimulation.

**Definition A.2** ($\pi$-bisimulation (Castro, 2020)). *Given a policy $\pi$, $\phi_{B,\pi}$ is a $\pi$-bisimulation if $\forall s_1, s_2 \in \mathcal{S}$ where $\phi_{B,\pi}(s_1) = \phi_{B,\pi}(s_2)$,*

$$
\sum_a \pi(a|s_1)R(s_1, a) = \sum_a \pi(a|s_2)R(s_2, a)
$$

$$
\sum_a \pi(a|s_1) \sum_{s' \in \phi_{B,\pi}^{-1}(z')} P(s'|s_1, a) = \sum_a \pi(a|s_2) \sum_{s' \in \phi_{B,\pi}^{-1}(z')} P(s'|s_2, a), \qquad \forall z' \in \mathcal{Z},
$$

*where $\mathcal{Z} := \phi_{B,\pi}(\mathcal{S})$ is the set of abstract states.*

However, $\pi$-bisimulation does not consider the state-action pair that is not visited under the policy $\pi$ (e.g., a state-action pair $(s, a)$ when $\pi(a|s) = 0$), whereas $Z^\pi$-irrelevance is defined on all the state-action pairs. Therefore, it is hard to build connection between them unless we also define $Z^\pi$-irrelevance on the state space (instead of the state-action space) in a similar way.

**Definition A.3** ($Z^\pi$-irrelevance on the state space). *Given $s \in \mathcal{S}$, we denote $\mathbb{Z}^\pi(s) := \sum_a \pi(a|s)Z^\pi(s, a)$. Given a policy $\pi$, $\phi$ is a $Z^\pi$-irrelevance if $\forall s_1, s_2 \in \mathcal{S}$ where $\phi(s_1) = \phi(s_2)$, $\mathbb{Z}^\pi(s_1) = \mathbb{Z}^\pi(s_2)$.*

Based on the above definitions, the following proposition indicates that such $Z^\pi$-irrelevance is coarser than $\pi$-bisimulation.

**Proposition A.2.** *Given a policy $\pi$ and $\phi_{B,\pi}$ to be the coarsest $\pi$-bisimulation, if $\forall s_1, s_2 \in \mathcal{S}$ where $\phi_{B,\pi}(s_1) = \phi_{B,\pi}(s_2)$, then $\mathbb{Z}^\pi(s_1) = \mathbb{Z}^\pi(s_2)$.*

*Proof.* Starting from $s_1$ and $s_2$ and following the policy $\pi$, the reward distribution and the state distribution over $\mathcal{Z}$ on each step are identical, which can be proved by induction. Then, we can conclude that $Z^\pi(s_1) \overset{D}{=} Z^\pi(s_2)$ and thus $\mathbb{Z}^\pi(s_1) = \mathbb{Z}^\pi(s_2)$. $\qquad\square$

### A.3 BOUND ON THE REPRESENTATION ERROR

For completeness, we restate Proposition 4.1.

**Proposition A.3.** *Given a policy $\pi$ and any $Z^\pi$-irrelevance $\phi : \mathcal{X} \to [N]$, there exists a function $Q : [N] \to \mathbb{R}$ such that $|Q(\phi(x)) - Q^\pi(x)| \leq \frac{R_{\max} - R_{\min}}{K}, \forall x \in \mathcal{X}$.*

*Proof.* Given a policy $\pi$ and a $Z^\pi$ irrelevance $\phi$, we can construct a $Q$ such that $Q(\phi(x)) = Q^\pi(x), \forall x \in \mathcal{X}$ in the following way: For all $x \in \mathcal{X}$, one by one, we assign $Q(z) \leftarrow Q^\pi(x)$, where $z = \phi(x)$. In this way, for any $x \in \mathcal{X}$ with $z = \phi(x)$, $Q(z) = Q^\pi(x')$ for some $x' \in \mathcal{X}$ such that $\mathbb{Z}^\pi(x') = \mathbb{Z}^\pi(x)$. This implies that $|Q(z) - Q^\pi(x)| = |Q^\pi(x') - Q^\pi(x)| \leq \frac{R_{\max} - R_{\min}}{K}$. This also applies to the optimal policy $\pi^*$. $\qquad\square$

## B PROOF

Notice that Corollary 4.1.1 is the asymptotic case ($n \to \infty$) for Theorem 4.1. We first provide a sketch proof for Corollary 4.1.1 which ignores sampling issues and thus more illustrative. Later, we provide the proof for Theorem 4.1 which mainly follows the techniques used in Misra et al. (2019).

### B.1 PROOF OF COROLLARY 4.1.1

Recall that Z-learning aims to solve the following optimization problem:

$$\min_{\phi \in \Phi_N, w \in \mathcal{W}_N} \mathcal{L}(\phi, w; \mathcal{D}) := \mathbb{E}_{(x_1, x_2, y) \sim \mathcal{D}} \left[ (w(\phi(x_1), \phi(x_2)) - y)^2 \right], \tag{5}$$

which can also be regarded as finding a compound predictor $f(\cdot, \cdot) := w(\phi(\cdot), \phi(\cdot))$ over the function class $\mathcal{F}_N := \{(x_1, x_2) \to w(\phi(x_1), \phi(x_2)) : w \in \mathcal{W}_N, \phi \in \Phi_N\}$.

For the first step, it is helpful to find out the Bayes optimal predictor $f^*$ when size of the dataset is infinite. We notice the fact that the Bayes optimal predictor for a square loss is the conditional mean, i.e., given a distribution $D$, the Bayes optimal predictor $f^* = arg\min_f \mathbb{E}_{(x,y) \sim D}[(f(x) - y)^2]$ satisfies $f^*(x') = \mathbb{E}_{(x,y) \sim D}[y \mid x = x']$. Using this property, we can obtain the Bayes optimal predictor over all the functions $\{\mathcal{X} \times \mathcal{X} \to [0,1]\}$ for our contrastive loss:

$$\begin{aligned} f^*(x_1, x_2) &= \mathbb{E}_{(x_1', x_2', y) \sim D}[y \mid x_1' = x_1, x_2' = x_2] \\ &= \mathbb{E}_{R_1 \sim Z^\pi(x_1), R_2 \sim Z^\pi(x_2)} \mathbb{I}[b(R_1) \neq b(R_2)] \\ &= 1 - \mathbb{Z}^\pi(x_1)^T \mathbb{Z}^\pi(x_2), \end{aligned} \tag{6}$$

where we use $D$ to denote the distribution from which each tuple in the dataset $\mathcal{D}$ is drawn.

To establish the theorem, we require that such an optimal predictor $f^*$ to be in the function class $\mathcal{F}_N$. Following a similar argument to Proposition 10 in Misra et al. (2019), it is not hard to show that using $N > N_{\pi, K}$ is sufficient for this realizability condition to hold.

**Corollary 4.1.1.** *The encoder $\hat{\phi}$ returned by Algorithm 1 with $n \to \infty$ is a $Z^\pi$-irrelevance, i.e., for any $x_1, x_2 \in \mathcal{X}$, $\mathbb{Z}^\pi(x_1) = \mathbb{Z}^\pi(x_2)$ if $\hat{\phi}(x_1) = \hat{\phi}(x_2)$.*

*Proof of Corollary 4.1.1.* Considering the asymptotic case (i.e., $n \to \infty$), we have $\hat{f} = f^*$ where $\hat{f}(\cdot, \cdot) := \hat{w}(\hat{\phi}(\cdot), \hat{\phi}(\cdot))$ and $\hat{w}$ and $\hat{\phi}$ is returned by Algorithm 1. If $\hat{\phi}(x_1) = \hat{\phi}(x_2)$, we have for any $x \in \mathcal{X}$,

$$1 - \mathbb{Z}^\pi(x_1)^T Z^\pi(x) = f^*(x_1, x) = \hat{f}(x_1, x) = \hat{f}(x_2, x) = f^*(x_2, x) = 1 - \mathbb{Z}^\pi(x_2)^T \mathbb{Z}^\pi(x).$$

We obtain $\mathbb{Z}^\pi(x_1) = \mathbb{Z}^\pi(x_2)$ by letting $x = x_1$ or $x = x_2$. $\qquad\square$

### B.2 Proof of Theorem 4.1

**Theorem 4.1.** *Given the encoder $\hat{\phi}$ returned by Algorithm 1, the following inequality holds with probability $1 - \delta$ and for any $x' \in \mathcal{X}$:*

$$
\mathbb{E}_{x_1 \sim d, x_2 \sim d}\left[\mathbb{I}[\hat{\phi}(x_1) = \hat{\phi}(x_2)] \left| \mathbb{Z}^\pi(x')^T \left(\mathbb{Z}^\pi(x_1) - \mathbb{Z}^\pi(x_2)\right)\right|\right]
$$
$$
\leq \sqrt{\frac{8N}{n}\left(3 + 4N^2 \ln n + 4\ln|\Phi_N| + 4\ln(\frac{2}{\delta})\right)}, \tag{7}
$$

*where $|\Phi_N|$ is the cardinality of encoder function class and $n$ is the size of the dataset.*

The theorem shows that 1) whenever $\hat{\phi}$ maps two state-actions $x_1, x_2$ to the same abstraction, $\mathbb{Z}^\pi(x_1) \approx \mathbb{Z}^\pi(x_2)$ up to an error proportional to $1/\sqrt{n}$ (ignoring the logarithm factor), and 2) when the difference of the return distributions (e.g., $\mathbb{Z}^\pi(x_1) - \mathbb{Z}^\pi(x_2)$) is large, the chance that two state-action pairs (e.g., $x_1$ and $x_2$) are assigned with the same encoding is small. However, since state-action pairs are sampled i.i.d. from the distribution $d$, the bound holds in an average sense instead of the worse-case sense.

The overview of the proof is as follows: Note that the theorem builds connection between the optimization problem defined in equation 5 and the learned encoder $\hat{\phi}$. To prove the theorem, we first find out the minimizer $f^*$ of the optimization problem when the number of samples is infinite (which was calculated in equation 6 in the previous subsection). Then, we bound the difference between the learned predictor $\hat{f}$ and $f^*$ in Lemma B.2 (which utilizes Lemma B.1) when the number of samples is finite. At last, utilizing the compound structure of $\hat{f}$, we can relate it to the encoder $\hat{\phi}$. Specifically, given $x_1, x_2 \in \mathcal{X}$ with $\hat{\phi}(x_1) = \hat{\phi}(x_2)$, we have $\hat{f}(x_1, x') = \hat{f}(x_2, x'), \forall x' \in \mathcal{X}$.

**Definition B.1** (Pointwise covering number). *Given a function class $\mathcal{G} : \mathcal{X} \to \mathbb{R}$, the pointwise covering number at scale $\epsilon$, $\mathcal{N}(\mathcal{G}, \epsilon)$, is the size of the set $V : \mathcal{X} \to \mathbb{R}$ such that $\forall g \in \mathcal{G}, \exists v \in V : \sup_{x \in \mathcal{X}} |g(x) - v(x)| < \epsilon$.*

**Lemma B.1.** *The logarithm of the pointwise covering number of the function class $\mathcal{F}_N$ satisfies $\ln \mathcal{N}(\mathcal{F}_N, \epsilon) \leq N^2 \ln(\frac{1}{\epsilon}) + \ln|\Phi_N|$.*

*Proof.* Recall that $\mathcal{F}_N := \{(x_1, x_2) \to w(\phi(x_1), \phi(x_2)) : w \in \mathcal{W}_N, \phi \in \Phi_N\}$, where $\mathcal{W}_N := \{[N] \times [N] \to [0, 1]\}$ and $\Phi_N := \{\mathcal{X} \to [N]\}$. Given $\epsilon > 0$, we discretize $[0, 1]$ to $Y := \{\epsilon, \cdots, \lceil\frac{1}{\epsilon}\rceil\epsilon\}$. Then, we define $W_N := \{[N] \times [N] \to Y\}$ and it is easy to see that $W_N$ is a covering of $\mathcal{W}_N$ with $|W_N| \leq (\frac{1}{\epsilon})^{N^2}$. Next, we observe that $F_N := \{(x_1, x_2) \to w(\phi(x_1), \phi(x_2)) : w \in W_N, \phi \in \Phi_N\}$ is a covering of $\mathcal{F}_N$ and $|F_N| = |\Phi_N||W_N|$. We complete the proof by combining the above results. $\square$

**Proposition B.1** (Proposition 12 in Misra et al. (2019)). *Consider a function class $\mathcal{G} : \mathcal{X} \to \mathbb{R}$ and $n$ samples $\{(x_i, y_i)\}_{i=1}^n$ drawn from $D$, where $x_i \in \mathcal{X}$ and $y_i \in [0, 1]$. The Bayes optimal function is $g^* = \arg\min_{g \in \mathcal{G}} \mathbb{E}_{(x,y) \sim D}\left[(g(x) - y)^2\right]$ and the empirical risk minimizer is $\hat{g} = \arg\min_{g \in \mathcal{G}} \frac{1}{n}\sum_{i=1}^n \left[(g(x_i) - y_i)^2\right]$. With probability at least $1 - \delta$ for a fixed $\delta \in (0, 1)$,*

$$
\mathcal{E}_{(x,y) \sim D}\left[(\hat{g}(x) - g^*(x))^2\right] \leq \inf_{\epsilon > 0}\left[6\epsilon + \frac{8\ln(2\mathcal{N}(\mathcal{G}, \epsilon)/\delta)}{n}\right],
$$

**Lemma B.2.** *Given the empirical risk minimizer $\hat{f}$ in Algorithm 1, we have*

$$
\mathbb{E}_{(x_1, x_2) \sim D}\left[(\hat{f}(x_1, x_2) - f^*(x_1, x_2))^2\right] \leq \Delta_{reg} = \frac{6}{n} + \frac{8}{n}\left(N^2 \ln n + \ln|\Phi_N| + \ln(\frac{2}{\delta})\right), \tag{8}
$$

*with probability at least $1 - \delta$.*

*Proof.* This directly follows from the combination of Lemma B.1 and Proposition B.1 by letting $\epsilon = \frac{1}{n}$. $\square$

*Proof of Theorem 4.1.* First, we denote $\mathcal{E}_i := \mathbb{I}[\hat{\phi}(x_1) = i = \hat{\phi}(x_2)]$ and $b_i := \mathbb{E}_{x \sim d}\left[\mathbb{I}[\hat{\phi}(x) = i]\right]$ to be the prior probability over the $i$-th abstraction. Then, for all $x' \in \mathcal{X}$, we have

$$\mathbb{E}_{x_1 \sim d, x_2 \sim d}\left[\mathbb{I}[\hat{\phi}(x_1) = i = \hat{\phi}(x_2)]\left|\mathbb{Z}^\pi(x')^T\left(Z^\pi(x_1) - Z^\pi(x_2)\right)\right|\right]$$

$$=\mathbb{E}_{x_1 \sim d, x_2 \sim d}\left[\mathcal{E}_i\left|f^*(x_1, x') - f^*(x_2, x')\right|\right]$$

$$\leq\mathbb{E}_{x_1 \sim d, x_2 \sim d}\left[\mathcal{E}_i\left(\left|f^*(x_1, x') - \hat{f}(x_1, x')\right| + \left|f^*(x_1, x') - \hat{f}(x_2, x')\right|\right)\right]$$

$$=\mathbb{E}_{x_1 \sim d, x_2 \sim d}\left[\mathcal{E}_i\left(\left|f^*(x_1, x') - \hat{f}(x_1, x')\right| + \left|f^*(x_1, x') - \hat{f}(x_1, x')\right|\right)\right]$$

$$\leq 2\mathbb{E}_{x_1 \sim d}\left[\mathbb{I}[\hat{\phi}(x_1) = i]\left|f^*(x_1, x') - \hat{f}(x_1, x')\right|\right]$$

$$\leq 2\sqrt{b_i \Delta_{\text{reg}}}.$$

The first step is obtained using Equation (6). The second step is the triangle inequality. The third step uses the fact that under event $\mathcal{E}_i$, we have $\hat{\phi}(x_1) = \hat{\phi}(x_2)$ and therefore $\hat{f}(x_1, x') = \hat{f}(x_2, x')$. In the fourth step, we drop the dependence on the other variable, and marginalize over $D_{\text{coup}}$. The last step is the Cauchy-Schwarz inequality.

At last, we complete the proof by summing over all the $i \in [N]$ and using the fact $\sum_{i=1}^N \sqrt{b_i} \leq \sqrt{N}$ and Lemma B.2. □

## C    IMPLEMENTATION DETAILS

### C.1    IMPLEMENTATION FOR ATARI GAMES

**Codebase.** For Atari games, we use the codebase from `https://github.com/Kaixhin/Rainbow` which is an implementation for data-efficient Rainbow (van Hasselt et al., 2019).

**Network architecture.** To implement our algorithm, we modify the architecture for the value network as shown in Figure 1 Left. In data-efficient Rainbow, the state embedding has a dimension of $576$. We maintain an action embedding for each action, which is a vector of the same dimension and treated as trainable parameters. Then, we generate the state-action embedding by conducting an element-wise product to the state embedding and the action embedding. This state-action embedding is shared with the auxiliary task and the main RL task. Afterwards, the value network outputs the return distribution for this state-action pair (noting that Rainbow uses a distributional RL algorithm, C51 (Bellemare et al., 2017)) instead of the return distributions of all actions for the input state as is in the original implementation.

**Hyperparameters.** We use exactly the same hyperparameters as those used in van Hasselt et al. (2019) and CURL (Srinivas et al., 2020) to quantify the gain brought by our auxiliary task and compare with CURL. We refer the readers to their paper for the detailed list of the hyperparameters.

**Balancing the auxiliary loss and the main RL loss.** Unlike CURL (or other previous work such as Jaderberg et al. (2016); Yarats et al. (2019)) that uses different/learned coefficients/learning rates for different games to balance the auxiliary task and the RL updates, our algorithm uses equal weight and learning rate for both the auxiliary task and the main RL task. This demonstrates the our auxiliary task is robust and does not need careful tuning for these hyperparameters compared with the previous work.

**Auxiliary loss.** Since the rewards in Atari games are sparse, we divide the segments such that all the state-action pairs within the same segment have the same return. This corresponds to the setting of Z-learning with $K \to \infty$ where the positive sample has exactly the same return with that of the anchor. Then, the auxiliary loss for each update is calculated as follows: First, we sample a batch of $64$ anchor state-action pairs from the prioritized replay memory. Then, for each state-action pair, we sample the corresponding positive pair (i.e., the state-action pair within the same segment as the anchor state-action pair) and the corresponding negative pair (randomly selected from the replay memory). The auxiliary loss is calculated on these samples with effectively $|\mathcal{D}| = 128$.

### C.2 IMPLEMENTATION FOR DMCONTROL SUITE

**Codebase.** We use SAC as the base RL algorithm and build our algorithm on top of the publicly released implementation from CURL (Srinivas et al., 2020).

**Network architecture.** Similarly, we modify the architecture for the critic network in SAC. In SAC, the state embedding has a dimension of 50. Since the actions are continuous vectors of dimension $d$ in the continuous control tasks of DMControl suite, we directly concatenate the action to the state embedding, resulting in a state-action embedding of size $50 + d$. Then, the critic network receives the state-action embedding as the input and outputs the Q value. The actor network receives the state embedding as the input and output the action selection distribution on the corresponding state. Note that, although our auxiliary loss is based on the state-action embedding, the state embedding used by the actor network is also trained by the auxiliary loss through back-propagation of the gradients.

**Hyperparameters.** We set the threshold for dividing the segments to $1.0$, i.e., when appending transitions to the replay buffer, we start a new segment when the cumulative reward within the last segment exceeds this threshold. The auxiliary loss and the hyperparameters to balance the auxiliary loss and the main RL loss are the same as those used for Atari games. Other hyperparameters we use are exactly the same as those in CURL implementation and we refer the readers to their paper for the details.

## D ADDITIONAL EXPERIMENT RESULTS

### D.1 MORE SEEDS RESULTS OF REPRESENTATION ANALYSIS

For clearness, we only show the result of representation analysis with a single seed in the main text. We add the results for multiple seeds here. The detailed description of analysis task can be found in the first paragraph in Section 5.2.

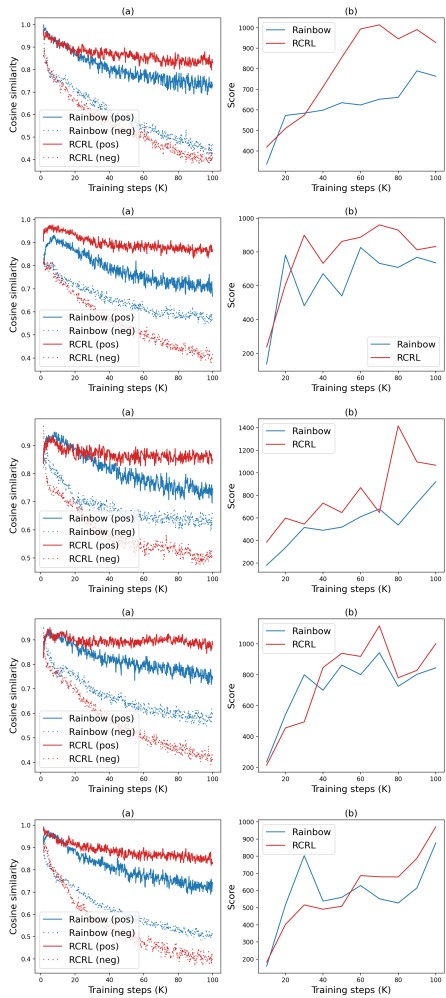

Figure 4: Analysis of the learned representation on *Alien* with five seeds. (a) The cosine similarity between the representations of the positive/negative state-action pair and the anchor during the training of Rainbow and RCRL. (b) The game scores of the two algorithms during the training.

## D.2 HIGH DATA REGIME RESULTS

To empirically study how applicable our model is to higher data regimes, we run the experiments on the first five Atari games (of Table 1) for 1.5 millon agent interactions. We show the evaluation results of both our algorithm and the rainbow baseline in Table 2. We can see that RCRL outperforms the ERainbow-sa baseline for 4 out of 5 games, which may imply that our auxiliary task has the potential to improve performance in the high-data regime.

| Game | ERainbow-sa (100k) | RCRL (100k) | ERainbow-sa (1.5M) | RCRL (1.5M) |
|------|--------------------|-------------|--------------------|-------------|
| Alien | 813.8 | 854.2 | 1721 | **1824** |
| Amidar | 154.2 | 157.7 | 398.8 | **454.5** |
| Assault | 576.2 | 569.6 | 572.5 | **757.9** |
| Asterix | 697 | 799 | **1370** | 1306.7 |
| Bank Heist | 96 | 107.2 | 257.3 | **550.7** |

Table 2: Scores of RCRL and ERainbow-sa on the first five Atari games.

