# OpenReview forum: "Return-Based Contrastive Representation Learning for Reinforcement  Learning"
_ICLR.cc/2021/Conference — ICLR 2021 Poster_

### Official Review · AnonReviewer3 · 2020-10-27
**Nice results, but focus on the low data regime should be clarified**

**Rating:** 7
**Confidence:** 3

**Review:**

**Summary**

The authors propose the inclusion of an auxiliary task for training an RL model, where the auxiliary task objective is to learn an abstraction of the state-action space that clusters (s,a) pairs according to their expected return.  The authors first describe a basic abstraction learning framework (Z-learning) followed by the extension to Deep RL as an auxiliary task (RCRL). The authors present results in Atari (discrete action) building on Rainbow, showing an improvement compared to baselines on median HNS in the low-data regime, and results on DMControl (continuous action) building on SAC, showing similar or improved performance compared to baselines.

**Quality**

Overall I found the approach and results to be interesting and moderately compelling. At first glance the improvement is surprising, given that model-free Deep RL already needs to abstract the state space on the basis of returns even without an auxiliary task. The key appears to be the focus on sample efficiency in the low-data regime, where the task seems to improve non-local value signal propagation compared to a bootstrapped algorithm (particularly on Atari, note that in the 100k regime the model-free algorithms have not yet learned to play Pong). Since it is not clear that the algorithm will generalize to more data (it's easy to imagine that the abstraction task will hinder performance when the base algorithms become more finely tuned), I would like to see more clarification of the goal throughout the paper (e.g. "In the low-data regime, our algorithm outperforms strong baselines on complex tasks in the DeepMind Control suite and Atari games" in the Abstract), as well as a reference to the focus in the Conclusions.

In the low-data regime, it's also critical to justify this approach compared to a model-based alternative. On Atari the authors compare to SimPLe, but MuZero would be a stronger baseline.

Besides the empirical results, the authors also nicely provide a description of the Z_\pi abstraction and an error bound.

**Clarity**

I was confused by the description of the positive/negative sampling procedure in 4.3 paragraph 2. Are segments temporally consecutive within a trajectory? If so, is it primarily a heuristic that they will "contain state-action pairs with similar returns" (i.e. couldn't a reward achieved mid-segment make this statement incorrect)? As I understand it, segmenting avoids the problem of determining bins on the return distribution a-priori, however it also seems like it will limit the agent's ability to cluster non-local (s,a) pairs with the same returns. It might also mean that the agent is learning to cluster temporally adjacent states in the underlying state space rather than similar returns.

**Originality**

The paper builds on existing work in the abstraction literature and auxiliary tasks for deep RL. The primary novel component is using a return-based auxiliary task.

The Z_\pi abstraction framework also appears to be novel, although its closely related to existing abstractions like Q_\pi abstraction.

**Significance**

The RCRL model itself improves on existing model-free approaches and can be easily incorporated into many model-free architectures, although it seems unlikely to beat a strong model-based baseline like MuZero in the low-data regime.

The description of the Z_\pi abstraction, and the exploration of return-based auxiliary tasks in general, could prove more significant in the long term.

**Pros:**
- The model improves performance in the low-data regime over existing model-free baselines
- The model can be easily added to many existing architectures
- Description and theoretical results on a new type of abstraction

**Cons:**
- The paper needs some more clarity around the focus on low-data / sample efficiency and how applicable the model is to higher data regimes
- Unclear if the segment-based sampling strategy is clustering (s,a) pairs with similar returns or just states that are nearby in the underlying state space
- The model seems unlikely to improve on a stronger model-based baseline in the low-data regime

---

> ### Author Response · Authors · 2020-11-19
> **Author Response to Reviewer 3**
>
> Thanks for your encouraging and constructive feedback! We address the main concerns below:
>
> #1. Q: The paper needs some more clarity around the focus on low-data / sample efficiency and how applicable the model is to higher data regimes.
>
> A: Thanks for the suggestion. We have clarified the focus on the low-data regime in the abstract and conclusion. However, when data is sufficient, the policy is fined tuned (approaching optimal) and the abstraction is learned. Theoretically in this situation, we can see that our auxiliary task is well aligned with the main task and therefore will not hinder the performance (cf. Proposition 4.1). Empirically, we will soon add results in high-data regimes on several Atari games to the appendix.
>
> #2. Q: Unclear if the segment-based sampling strategy is clustering $(s,a)$ pairs with similar returns or just states that are nearby in the underlying state space.
>
> A: We did not make it clear and we have updated the description for the segmentation procedure in the revised version of our paper (cf. the second paragraph in Section 4.3). The segmentation procedure is as follows: In Atari games, we create a new segment once the agent receives a non-zero reward. Namely, there is no reward achieved mid-segment. In DMControl tasks, we first prescribe a threshold and then create a new segment once the cumulative reward within the current segment exceeds this threshold. In this way, we are essentially clustering state-action pairs with similar returns instead of simply the nearby state-action pairs. As an example, consider a transition from one state-action pair ($x_1$) to another ($x_2$) with a non-zero reward. Although $x_1$ and $x_2$ are temporally adjacent, we do not cluster them together. Actually, the pair ($x_1$, $x_2$) can even be considered as a negative sample. Since $x_1$ and $x_2$ have different returns, it is possible that our algorithm may sample ($x_1$, $x_2$) as a negative sample.
>
> #3. Q: The model seems unlikely to improve on a stronger model-based baseline in the low-data regime.
>
> A: Indeed, current RCRL (that is combined with base model-free algorithms) does not outperform stronger model-based algorithms such as MuZero. We have modified relevant statements in the revised version. As future work, it is interesting to combine our auxiliary task with model-based algorithms.

---

> > ### Comment · AnonReviewer3 · 2020-11-24
> > **Response to Authors**
> >
> > Thank you for the clear response and the helpful highlighted version of the updated text. Two additional questions about the positive-negative sampling approach (these would be beneficial to comment on in the final text):
> >
> > (1) Both the Atari and DMControl segmenting approaches seem tailored to the reward distributions of the tasks; can we expect that these or other approaches will be robust to many types of tasks with varying reward sparsity?
> >
> > (2) Say that two (s,a) pairs have exactly the same return distribution but are never experienced in the same trajectory. It seems that these (s,a) pairs should be aggregated in an optimal Z^\pi abstraction, but (generally speaking) they will not be positive sampled together. Is this correct, and do you think it limits the RCRL algorithm?

---

> > > ### Author Response · Authors · 2020-11-24
> > > **Thanks for your reply**
> > >
> > > Thanks for your reply. We answer the questions below:
> > >  1) Although described separately, the segmenting methods for Atari and DMControl are essentially the same but only different thresholds are selected. In Atari, we set the threshold to be 1. (Note that the rewards in Atari are discretized to $\\{-1,0,1\\}$.) In terms of determining the threshold, the principle is to choose a threshold such that there is a reasonable number of segments in one trajectory (e.g., 10~20). We believe our method should work for different tasks with varying reward sparsity as long as a proper threshold is selected.
> > >
> > >  2) Yes, this is correct. However, it may not limit the performance since it leads to an abstraction that is finer than $Z^\pi$ abstraction. Note that a finer abstraction is still able to represent the Q values.
> > >
> > > Besides, we would like to update some results in high data regime.  We show the results on the first five Atari games (in alphabetical order) for 1.5 million agent interactions in the table below (as well as in Appendix D.2.), and we will add the results for remaining games to the camera-ready version. We can see that RCRL outperforms the rainbow baseline in 4 out of 5 games, which may imply that our auxiliary task has the potential to improve performance in the high data regime.
> > >
> > > | Game | ERainbow-sa (100k) | RCRL (100k) | ERainbow-sa (1.5M) | RCRL (1.5M) |
> > > | :----: | :----: | :----: | :----: | :----: |
> > > | Alien| 813.8| 854.2| 1721| **1824** |
> > > | Amidar  | 154.2  | 157.7 | 398.8 | **454.5**
> > > | Assault  | 576.2 | 569.6 | 572.5 | **757.9**
> > > | Asterix  | 697     | 799    | **1370**  | 1306.7
> > > | Bank Heist | 96 | 107.2 | 257.3 | **550.7**

---

> > > > ### Comment · AnonReviewer3 · 2020-11-24
> > > > **Reply to Authors**
> > > >
> > > > Very interesting, thank you, it's good to see that the results extend to high-data regimes. I will reassess my rating as a result.

---

> > > > ### Comment · AnonReviewer1 · 2020-11-24
> > > > **Impressive results in high data regime.**
> > > >
> > > > It is good to know that the proposed approach may not just be "speeding up convergence", but may actually be increasing the maximum attainable returns.
> > > >
> > > > Are the results obtained from multiple seeds?

---

> > > > > ### Author Response · Authors · 2020-11-24
> > > > > **Thanks for your reply**
> > > > >
> > > > > Yes, these results are obtained from five random seeds (the same number of seeds with Table 1).

---

### Official Review · AnonReviewer1 · 2020-10-28
**Relevant topic with novel formulation**

**Rating:** 6
**Confidence:** 3

**Review:**

## Return-Based Contrastive Representation Learning for Reinforcement Learning
### Summary
The authors propose Return-based Contrastive Representation Learning (RCRL), a contrastive auxiliary learning task that guides the feature network to encode representation relevant to the task rewards. The experiment results show that RCRL helps improve two commonly used RL algorithm (RAINBOW and SAC) in low data regime. Additionally, RCRL can also be used in combination with other auxiliary tasks to boost performance.

Overall, the paper is well-written, the topic is relevant to the field and the approach is novel.

### Strength
- Theoretically-backed.
- The topic of representation learning is pretty relevant to the field now.
- The learned representation is "task-relevant", and therefore can achieve higher performance compared to other representation learning methods.

### Weakness
- The reliance on environment returns make the approach pretty susceptible to poorly or sparsely defined rewards. Specifically:
  1. The auxiliary loss does not work in sparse reward environments.
  2. In the task with dense but deceptive rewards, the representation may be biased toward representation that is not helpful in the long run.
- The improvement in continuous control tasks seem to be really marginal. Why is that?
- The learned representation may not be very general due to its reliance on return signals. Certainly, it can help achieve better performance when we only focus on a single task with a well-defined reward function. Yet, the representation may not be as useful when we considered some practical real-world settings that require policy adaptation and transfer learning.

---

> ### Author Response · Authors · 2020-11-19
> **Author Response to Reviewer 1**
>
> Thanks for your encouraging and constructive feedback! We address the main concerns below:
>
> #1. Q: The reliance on environment returns make the approach pretty susceptible to poorly or sparsely defined rewards.
>
> A:  It is true that our approach may not work well for the poorly or sparsely defined rewards. However, it is generally believed that such tasks are hard and may require additional measures. For example, we may first perform reward shaping or credit assignment and then apply our approach.
>
> #2. Q: The improvement in continuous control tasks seem to be marginal. Why is that?
>
> A: Different from Atari games where typical implementations often clip rewards to [-1, 0, 1] (three discrete values) in the training, the reward functions for different DMControl tasks have different ranges. We use the same set of hyperparameters for all DMControl tasks which may not be the best for each task. Tuning these hyperparameters (such as the threshold for segmentation) or scaling the reward per task may further boost the performance.
>
> #3. Q: The learned representation may not be very general due to its reliance on return signals. Certainly, it can help achieve better performance when we only focus on a single task with a well-defined reward function. Yet, the representation may not be as useful when we considered some practical real-world settings that require policy adaptation and transfer learning.
>
> A:  Thanks for this insight. The reliance on specific return signals may make our approach not work well in the policy adaptation or transfer learning setting. For such settings, we may apply our method to more than one reward function (either provided in advance or automatically generated). The resultant representation should be more robust and may be more suitable for adaptation or transfer. Similar approaches can also be found in [1] and [2].
>
> [1] Jaderberg, Max, et al. "Reinforcement learning with unsupervised auxiliary tasks." arXiv preprint arXiv:1611.05397 (2016).
>
> [2] Veeriah V, Hessel M, Xu Z, et al. "Discovery of useful questions as auxiliary tasks." Advances in Neural Information Processing Systems. 2019: 9310-9321.

---

> > ### Comment · AnonReviewer1 · 2020-11-23
> > **Thanks for addressing the raised questions**
> >
> > Thanks for the responses. Just one more question: about hyperparameter tuning: "the reward functions for different DMControl tasks have different ranges..."
> > This kind of corresponds to my original point: if the reward function is not carefully designed, then extra works have to be done to make the method work. It would be nice if there are some principal ways that you can use for determining those hyperparameters.
> >
> > Overall, I like the idea of this approach, and the results look promising in some settings. But the approach does have room for improvement (e.g., generalizability of the learned representation, need well-defined and well-ranged reward functions). Therefore, I tend to stick with my original rating for now. I will keep track of the discussions and your responses to other reviewers.

---

> > > ### Author Response · Authors · 2020-11-24
> > > **Thanks for your reply**
> > >
> > > Thanks for your reply. Indeed, our hyperparameter (the threshold) is sensitive to the reward range in different tasks. For robustness to tasks with different reward ranges, we can normalize the reward range (e.g., using a standard scaler which removes the mean and scales to unit variance with running mean and variance). Similar normalization tricks are widely used to normalize observations in many RL implementations. In terms of determining the threshold, a principal way is to choose a threshold such that there is a reasonable number of segments in one trajectory (e.g., 10~20).

---

### Official Review · AnonReviewer4 · 2020-10-28
**Review of "Return-Based Contrastive Representation Learning for Reinforcement Learning"**

**Rating:** 7
**Confidence:** 4

**Review:**


Summary:

The authors present a contrastive auxiliary loss based upon state-action returns.

They introduce an abstraction over state-action pairs and divide the space of state-action returns into K bins over which the Z function is defined where Z(s,a) is distributed over K-dimensional vectors.  Given an encoding function, phi, and an input x, Z-irrelevance is defined as phi(x_1) = phi(x_2) when Z(x_1) = Z(x_2) which motivates the objective for Z-Learning: to classify state-action pairs with similar returns (within bounds) to be similar.  From this a contrastive loss can be defined (Return-based Contrastive RL, RCRL) where class labels are determined by Z-irrelevance sets encouraging state-action encodings to be similar when the returns are.  In the limit Z becomes the RL state-action value function Q.

The authors evaluate their approach on Atari (discrete actions) and the DeepMind Control Suite (continuous actions) across both model-free and model based RL algorithms against and in combination with other auxilliary losses including CURL (Srinivas et al. 2020).

Strengths & Weaknesses:

Auxiliary losses have become an important component in RL for developing stable agents that can generalize well and form good representations.  In particular, contrastive losses have come into increasing use with growing literature around these methods and so I believe the domain area of this paper is relevant and of interest.  The authors do a good job of covering the recent developments of background literature in their related work section and grounding their approach with recent efforts undertaken in RL auxiliary losses, contrastive learning approaches and state abstraction/representation learning literature.

The approach is overall novel as many contrastive learning methods are defined against input data or downstream representations, whereas this work derives it's data from RL returns and creates a link between the representational landscape of the observations and actions and broad outcomes as they are valuable to an agent.  As the author's have framed the problem, I believe this approach is more powerful and also more tractable than something like reward prediction.  Intuitively the formulation seems solid to me since we often would like to understand not only when we're in a good state and taking a useful action but also, in general, what kind of properties state-action pairs with similar returns should have.  The authors do note that this may be learnable by temporal difference updates alone however, this approach aims to directly encourage the learning of this relationship and decouple it from the RL algorithm (where perhaps other things may be focused on such as planning etc.).

One shortfall of this approach could be the available data itself as you'd rely on the policy to provide you with good samples for RCRL.  The authors indicate that they segment trajectories to ensure better quality positive and negative samples for learning however, it could be made clearer how much of a problem this can be.    This approach could possibly be combined with a self-supervised approach to alleviate these types of concerns.   It would also be nice to know the additional computational burden of RCRL and how this compares to other auxiliary losses.

The experiments on Atari & Control look solid and demonstrate that this method attained a stronger score both alone and when combined with CURL and good top performance on DeepMind control suite tasks when compared again to CURL and pixelSAC.  It might have been nice to see more comparisons or combinations with other contrastive methods that have had some success in  learning visual representations (SimCLR: Chen et al. 2020, BYOL: Grill et al. 2020).  The similarity analysis also provided some nice insight into the inductive bias induced by RCRL.

Overall, the paper is well written and has a clear layout.  The authors provide clear algorithms and figures and the content flows well from section to section.


Recommendation:

I believe that this is a promising and very active area of research and that this work makes the case for a solid new approach and a set of encouraging results to back it up.

---

> ### Author Response · Authors · 2020-11-19
> **Author Response to Reviewer 4**
>
> Thanks for your encouraging and constructive feedback! We address the main concerns below:
>
> #1 Q: One shortfall of this approach could be the available data itself as you'd rely on the policy to provide you with good samples for RCRL.
>
> A: To obtain a $Z^\pi$-irrelevance abstraction, we indeed need the samples (data) from the policy $\pi$. Fortunately, these data are always available during the policy optimization process. Besides, as is discussed in [1], such on-policy abstractions may be more efficient by focusing only on the behavior of interest, rather than worst-case scenarios (induced by other policies).
>
> #2 Q: The authors indicate that they segment trajectories to ensure better quality positive and negative samples for learning however, it could be made clearer how much of a problem this can be.
>
> A: Thanks for your suggestion. If we directly use the sampling procedure of Z-learning, the ratio of positive and negative samples is around 2:8 (in the game Alien). With the segment based method, we make the ratio to be 1:1 (and thus balanced), which shows good empirical performance.
>
> #3 Q: It would also be nice to know the additional computational burden of RCRL and how this compares to other auxiliary losses.
>
> A: The additional computational burden comes from the optimization of our contrastive loss (i.e., Equation 1) and the additional sampling procedure. Empirically, our auxiliary task costs 21% additional training time on top of Rainbow whereas CURL costs 30%. Compared with CURL, our contrastive loss is more efficient computationally, since our loss does not take all other samples in a batch as negative samples.
>
> #4 Q: It might have been nice to see more comparisons or combinations with other contrastive methods that have had some success in learning visual representations (SimCLR: Chen et al. 2020, BYOL: Grill et al. 2020).
>
> A: Yes, our design of return-based auxiliary loss is orthogonal to the specific techniques of contrastive learning. We believe that combining the advantages of more advanced contrastive learning methods with return-based auxiliary loss is an exciting direction for further research.
>
> [1] Castro, Pablo Samuel. ``Scalable methods for computing state similarity in deterministic markov decision processes." Proceedings of the AAAI Conference on Artificial Intelligence. Vol. 34. No. 06. 2020.

---

### Official Review · AnonReviewer2 · 2020-10-29
**Official blind review**

**Rating:** 6
**Confidence:** 4

**Review:**

This is an interesting paper that proposes abstractions based on return distribution similarities to be used as auxiliary tasks to aid in learning. The idea is quite promising and I think could open up new avenues for future research, but it does not appear to me to be ready for publication yet. In particular, although the authors claim to be distinguishing based on returns, many of the design decisions seem to implicitly assume determinism, or near-determinism (in particular, see point 3 and 10 below).
The theoretical results seem interesting, but I have some questions on their validity and clarity (see point 7 and 8 below).
The practical algorithm has a few issues that need clarification (see points 3, 4, 9, 11, and 12 below).

1. At the bottom of page 2, the authors write "we are the first to leverage return to construct a contrastive auxiliary task for speeding up the main RL task." This is not quite true, see [Zhang, A.; McAllister, R.; Calandra, R.; Gal, Y.; and Levine, S. 2020. Learning Invariant Representations for Reinforcement Learning without Reconstruction". arXiv preprint arXiv:2006.10742 .].
2. In section 2.2 you should cite [Taylor, J.; Precup, D.; and Panagaden, P. 2009. Bounding performance loss in approximate MDP homomorphisms. In Advances in Neural Information Processing Systems, 1649–1656.]
3. In line 6 of Algorithm 1, $y = \mathbb{I}[b(R_1) \ne b(R_2)]$ seems problematic for stochastic returns. In particular, if the number of bins is very large and the returns have wide variance, $y$ will almost always be zero.
4. In line 8 of Algorithm 1 is $\hat{w}$ also learned?
5. In the last sentence of the first paragraph of section 4.1, rather than comparing against regular bisimulation it seems more appropriate to compare to $\pi$-bisimulation from [Castro, P. S. 2020. Scalable methods for computing state similarity in deterministic Markov Decision Processes. In Proceedings of the AAAI Conference on Artificial Intelligence.].
6. In equation (1), the minimization appears over $f$, but $f$ doesn't appear on the RHS. Shouldn't the minimization be over $w$?
7. In Theorem 4.1 it's not clear what role $x_1$ plays in the result. Why is this third state necessary? It does not seem to show up in the proof in the appendix. Further, the proof in the appendix could do with some elaboration, as it's not completely clear how lemmas B.1 and B.2 result in the proof of Theorem 4.1. It would be better if the authors restate the theorem statement in the appendix and were more explicit about the connections. There are no page limits for the appendix.
8. How do we get a sense for how big $|\Phi_N|$ is? Couldn't it be as large as $N^{\mathcal{|X|}}$?
9. Typically auxiliary losses are combined with the main loss into a single loss, but in Algorithm 2 you seem to be updating them sequentially. Why? Does the order of update matter?
10. In Figure 1, on the left, the purple rectangle says "Value net". If so, are you really learning a distribution?
11. In Figure 2, why do some algorithms only have triangles and not learning curves? Also, it seems from that figure that "State SAC (skyline)" seems to outperform all others, including RCRL.
12. Is Figure 3 over a single seed?
13. In Definition A.1, bisimulation was _not_ introduced in (Jiang, 2018), it was introduced in [Givan, R., Dean, T., & Greig, M. (2003). Equivalence notions and model minimization in markov decision processes. Artificial Intelligence, 147, 163–223].


Minor comments:
1. It would be helpful if the authors specify how to pronounce RCRL. While reading the paper I was pronouncing it [like this](https://youtu.be/dQw4w9WgXcQ).
2. In the second-to-last sentence on page 1, should read "while ignor**ing** return-irrelevant features."

---

> ### Author Response · Authors · 2020-11-19
> **Author Response to Reviewer 2**
>
> Thanks for your encouraging and constructive feedback! We address the main concerns below:
>
> #1 The paper (Zhang et al 2020) uses bisimulation metrics for representation learning where instant reward is leveraged, whereas our paper leverages long-term return. Moreover, their paper does not use contrastive learning, whereas our paper uses contrastive learning. Thanks for your suggestion and we have added the paper to our related work in the updated version.
>
> #2 Thanks for your suggestion and we have cited the paper in our revised version.
>
> #3 Our method is actually designed for stochastic returns. Indeed, as you pointed out, more samples are needed to distinguish between two different return distributions, when the variance of returns and the number of bins are large. However, the number of bins ($K$) is a preset hyperparameter, and we choose a proper number such that the ratio of positive labels ($y=0$) is not vanishing. In practice, we partition the return values in a trajectory by dividing it into segments and thus do not choose $K$ explicitly. Typically, the number of segments and the corresponding $K$ value are approximately $6\sim 20$. To further balance the numbers of positive/negative samples, we adopt a sampling procedure based on the segmented trajectories to sample positive and negative pairs at a ratio of 1:1. In our sampling procedure, we use the samples within one segment as positive pairs to generate more positive pairs. Please find more details in the second paragraph in Section 4.3.
>
> #4 Yes, both $\hat{w}$ and $\hat{\phi}$ are learnable. The learnable function $f$ is composed of the two learnable components, $\hat{w}$ and $\hat{\phi}$. Therefore, the minimization over $f$ is equivalent to the minimization over $\hat{w}$ and $\hat{\phi}$. We have updated the paper to make it clear.
>
> #5 Yes, it is more appropriate to compare with $\pi$-bisimulation. However, $\pi$-bisimulation is defined over the state space instead of the state-action space, where our $Z^{\pi}$-irrelevance is defined over the state-action space. For a fair comparison, we additionally define $Z^\pi$-irrelevance over the state space, and the result shows that  $Z^\pi$-irrelevance is coarser than $\pi$-bisimulation. See the details in Appendix A.2 in our revised version.
>
> #6 Please refer to #4.
>
> #7 Thanks for your advice. The third state-action pair (originally denoted as $x_1$ and currently denoted as $x'$) in the expectation is not necessary. We have updated the theorem (and updated the proof accordingly) and now it holds for any $x'$ and is stronger than the previous theorem. We also restated the theorem statement and explained more about the connections between the theorem and the lemmas in Appendix B.2.
>
> #8 $|\Phi_N|$ is the size of the encoder class which is upper bounded by $N^{|\mathcal{X}|}$. In practice, we use a deep encoder that is generalizable across different state-action pairs. In this case, this term is effectively much smaller than $N^{|\mathcal{X}|}$, which is a general assumption for deep RL models. We add the discussion on $|\Phi_N|$ and $N^{|\mathcal{X}|}$ in the revised version of our paper.
>
> #9 We do not update them sequentially, and we combine the auxiliary loss with the main RL loss into a single loss.
> We did not make it clear and we have updated Algorithm 2 in the revised version.
>
> #10 Since we use Rainbow as the base RL algorithm and Rainbow uses C51 (which is a distributional RL algorithm), the value network learns to approximate the distribution of the return. This part is relatively independent of our auxiliary task. We modify the figure to avoid such confusion.
>
> #11 For these baselines, we use the reported results in the CURL paper (Table 2). However, only the points upon 100K and 500K are available, so we plot them in the figure using triangles. For CURL, we run their official open-source code to get the learning curve. State SAC uses the low-dimensional RAM state as the input, while others use the high-dimensional image as the input. Therefore, the tasks for State SAC are different from others and much easier. It is an unfair comparison, so we called it skyline. We can see that the performance of RCRL is comparable even to this skyline.
>
> #12 For clearness, we only showed the result with a single seed. We have added the results for multiple seeds in Appendix D in the updated version.
>
> #13 Thanks for your suggestion. We have revised the citation.
>
> Minor #1 Without a better idea to pronounce, we suggest pronouncing as it is (R-C-R-L). Another way is to read the algorithm as ReCoRL, pronounced as |re'kəʊɜl|. But ReCoRL is not as simple as RCRL in terms of writing. Thanks for the good song.
>
> Minor #2 We have fixed this grammar error in the revised version.

---

> > ### Comment · AnonReviewer2 · 2020-11-20
> > **Highlight changes?**
> >
> > Thanks for your response. In order to facilitate our reassessment of the paper, would it be possible to highlight the changes made to the draft by, for instance, making the text color of the changes blue (or some color other than black)?

---

> > > ### Author Response · Authors · 2020-11-21
> > > **Response**
> > >
> > > Thanks for your attention. You can find the highlighted version in Supplementary Material (in the zip file).

---

> > > > ### Comment · AnonReviewer2 · 2020-11-24
> > > > **Thanks!**
> > > >
> > > > Thanks for your responses and for highlighting the changes!

---

### Author Response · Authors · 2020-11-19
**Rebuttal Revision Has Been Posted**

Dear Reviewers, AC, PC, and Readers,

According to the comments of reviewers, we have revised and uploaded the paper. For clearness, we also uploaded the full paper with appendices to supplementary material where changes are highlighted in blue.

Thanks for your attention,

Paper 2378 Authors

---

### Decision · Program_Chairs · 2021-01-07
**Final Decision**

**Decision:**

Accept (Poster)

**Comment:**

RCRL is return-based contrastive learning for reinforcement learning, where the label is whether two samples belong to the same return bin. The reviewers found this to be a well executed paper with good theoretical and experimental results.